# Detection of Bacterial α-l-Fucosidases with an *Ortho*-Quinone Methide-Based Probe and Mapping of the Probe-Protein Adducts

**DOI:** 10.3390/molecules27051615

**Published:** 2022-02-28

**Authors:** Yvette M. C. A. Luijkx, Anniek J. Henselijn, Gerlof P. Bosman, Dario A. T. Cramer, Koen C. A. P. Giesbers, Esther M. van ‘t Veld, Geert-Jan Boons, Albert J. R. Heck, Karli R. Reiding, Karin Strijbis, Tom Wennekes

**Affiliations:** 1Department Chemical Biology and Drug Discovery, Utrecht Institute for Pharmaceutical Sciences, Bijvoet Center for Biomolecular Research, Utrecht University, Universiteitswg 99, 3584 CG Utrecht, The Netherlands; yvetteluijkx@gmail.com (Y.M.C.A.L.); a.j.henselijn@lic.leidenuniv.nl (A.J.H.); g.p.bosman@uu.nl (G.P.B.); g.j.p.h.boons@uu.nl (G.-J.B.); 2Department Biomolecular Mass Spectrometry and Proteomics, Utrecht Institute for Pharmaceutical Sciences, Bijvoet Center for Biomolecular Research, Utrecht University, 3584 CH Utrecht, The Netherlands; d.a.t.cramer@uu.nl (D.A.T.C.); a.j.r.heck@uu.nl (A.J.R.H.); k.r.reiding@uu.nl (K.R.R.); 3Netherlands Proteomics Centre, 3584 CH Utrecht, The Netherlands; 4Department Biomolecular Health Sciences, Division Infectious Diseases and Immunology, Faculty of Veterinary Medicine, Utrecht University, Yalelaan 1, 3584 CL Utrecht, The Netherlands; c.a.p.giesbers@uu.nl; 5Department Biomolecular Health Sciences, Centre for Cell Imaging, Faculty of Veterinary Medicine, Utrecht University, Yalelaan 1, 3584 CL Utrecht, The Netherlands; e.m.vantveld@uu.nl; 6Complex Carbohydrate Research Center, University of Georgia, Athens, GA 30602, USA

**Keywords:** quinone methide, probe, proteomics, glycosidase, bacteria, fucose

## Abstract

Fucosidases are associated with several pathological conditions and play an important role in the health of the human gut. For example, fucosidases have been shown to be indicators and/or involved in hepatocellular carcinoma, breast cancer, and helicobacter pylori infections. A prerequisite for the detection and profiling of fucosidases is the formation of a specific covalent linkage between the enzyme of interest and the activity-based probe (ABP). The most commonly used fucosidase ABPs are limited to only one of the classes of fucosidases, the retaining fucosidases. New approaches are needed that allow for the detection of the second class of fucosidases, the inverting type. Here, we report an *ortho*-quinone methide-based probe with an azide mini-tag that selectively labels both retaining and inverting bacterial α-l-fucosidases. Mass spectrometry-based intact protein and sequence analysis of a probe-labeled bacterial fucosidase revealed almost exclusive single labeling at two specific tryptophan residues outside of the active site. Furthermore, the probe could detect and image extracellular fucosidase activity on the surface of live bacteria.

## 1. Introduction

α-l-Fucosidases are enzymes capable of catalyzing the hydrolytic removal of terminal l-fucose residues from glycoconjugates. It is known that an abnormal increase in α-l-fucosidase activity in humans is associated with several pathological conditions [1,2,3,4]. Furthermore, bacterial α-l-fucosidases are key enzymes for the degradation and metabolism of intestinal mucin *O*-glycans by gut microbes. This crucial family of enzymes thereby contributes to the composition of the gut microbiota and influences our health and disease [5]. We and others recently reported that fucosidases of two *Bacteroides* species from the human gut microbiota induced upregulation of growth and invasive properties of pathogenic *Campylobacter jejuni* strains [6,7]. A growing number of studies are implicating bacterial fucosidases in the host–microbe interplay in the intestine, microbiota cross-feeding, and colonization resistance [3,5,8,9,10].

α-l-Fucosidases employ one of two mechanisms to catalyze the hydrolysis of terminal α-fucosidic linkages, and these result in either overall retention or inversion of the anomeric configuration. Fucosidases employing a double displacement (Koshland) retaining mechanism belong to the glycoside hydrolase family 29 (GH29), whereas those employing an inverting mechanism belong to the glycoside hydrolase family 95 (GH95) [11,12]. Given their importance and the vast amounts of fucosidases in the human gut microbiome, several attempts have been made to develop ABPs to elucidate their individual functions [13,14,15,16]. However, the majority of ABPs developed against fucosidases exploit the double replacement retaining mechanism of GH29 fucosidases and are thus incapable of targeting GH95 fucosidases [13,14,15,16], thereby highlighting the need to expand the molecular toolbox to study all known and still to be discovered bacterial α-l-fucosidases.

Quinone methide-based probes are a type of ABP that would allow the labeling of both GH29 and GH95 fucosidases [17,18]. Upon α-l-fucosidase cleavage of the glycosidic bond in such a probe, the liberated *ortho*-fluoromethyl-phenolate aglycone undergoes rapid 1,4-elimination through expulsion of a fluoride ion to generate a highly electrophilic *ortho*-quinone methide. This reactive intermediate quickly alkylates a nearby nucleophile, which results in the formation of a covalently probe-labeled enzyme adduct (Figure 1A). This type of probe has already been shown to be capable of labeling β-galactosidases, neuraminidases of *V. cholera*, and human GH29 α-l-fucosidase enzymes [4,17,19,20]. One of the challenges in the application of quinone methide-based probes is the possibility for off-target labeling of nucleophiles in nearby biomolecules due to diffusion of the *ortho*-quinone methide with its relatively long lifetime. However, this property also makes these probes an interesting candidate for imaging of enzyme activity as the probes do not react with the active site, which would lead to inactivation of the enzyme. Instead, the possibility for repeated hydrolysis of new copies of the probe could lead to the accumulation of multiple adducts near the active site. This strategy with multiple adducts has previously been applied for signal amplification in imaging studies. [21,22]

A quinone methide-based probe to detect fucosidase activity has previously been developed by Hsu et al. [19]. However, the application of this probe is limited because of the bulky hydrophobic reporter group that might prevent the probe from accessing the active site of some α-l-fucosidases in gut microbiota [23]. Instead, the installation of a mini-tag, such as an azide or an alkyne, could overcome this limitation and make the probe more flexible in the detection of labeling. Thus, to enable effective in vivo imaging of both GH29 and GH95 bacterial fucosidases, we developed a more compact and flexible version of the Hsu et al. probe by the incorporation of an azide mini-tag [19]. Conjugation of a reporter group to the probe following labeling of the enzyme target was accomplished by the azide mini-tag reacting via Huisgen’s 1,3-dipolar cycloaddition with a variety of alkyne-bearing reporter groups (Figure 1B) [19].

Herein, we report the synthesis of a quinone methide-based ABP that can directly confer the activity of α-L-fucosidases. AH062 shows effective and selective labeling of both recombinant GH29 and GH95 fucosidases. We used *Bacteroides fragilis* as a model bacterium to investigate imaging properties of the fucosidase-activated AH062 and show specific extracellular fucosidase activity. Furthermore, we used the recombinantly expressed fucosidases from *B. fragilis* and *Thermotoga maritima* to investigate, by MS analysis, which are the preferred amino acid residues modified by AH062.

## 2. Results and Discussion

We synthesized the self-immobilizing fucosidase probe AH062 in eight steps by adapting the synthetic route previously published by Hsu et al. (Figure 2). To determine the labeling efficiency of AH062, we incubated a recombinant *B. fragilis* fucosidase (BfFucH, GH29 fucosidase) at various concentrations (4-0.25 µg corresponding to 60-960 nM) with AH062 (350 µM) for 2 h at 37 °C. The resulting mixture was analyzed by SDS-PAGE followed by Western blotting. Detection was achieved by a Cu(I)-catalyzed click reaction on the blot surface with alkyne-biotin followed by incubation with anti-biotin HRP-linked antibody and detection using an enhanced chemiluminescence (ECL) substrate. The probe could detect the BfFucH enzyme at concentrations as low as 0.25 µg (60 nM) (Figure 1A).

AH062 was also capable of distinguishing the active from the inhibited enzyme (prepared by preincubation with the competitive inhibitor l-fuconojirimycin (FNJ)) [16]. Generally, a concern with quinone-methide based probes is their labeling selectivity, due to diffusion of the *ortho*-quinone methide trapping unit away from the targeted enzyme. To determine the labeling selectivity, we examined the ability of AH062 to specifically target fucosidases in the presence of a second protein. BfFucH samples were spiked with an equimolar concentration of transferrin (TF) and incubated with the probe. TF is a glycoprotein of 79.5 kDa and can therefore be clearly separated on gel from the ~50 kDa BfFucH. AH062 showed clear signals for BfFucH, but no chemiluminescent signal was detected for TF (Figure 1B). These results suggest that AH062 is specifically reactive in or near the active site of the fucosidase. Furthermore, treatment of TF with AH062 in the absence of active BfFucH did not give any detectable signal.

To assess the ability of AH062 to visualize catalytically active inverting GH95 fucosidases, we expressed the known GH95 fucosidase AfcA from *Bifidobacterium bifidum*. AH062 was incubated with five different concentrations of AfcA (8–3.7 µg), and the probe detected the enzyme in a concentration-dependent manner (Figure 1C). AH062 labeled as little as 4.8 µg of AfcA (Figure 1C). To our knowledge, this is the first example of an ABP capable of labeling of an inverting fucosidase (GH95).

We next performed mass spectrometric studies to determine the attachment sites of the novel quinone methide probe onto the fucosidase enzyme. After incubation of AH062 for 2 h at 37 °C with BfFucH, intact LC-MS profiles from labeled and untreated samples were compared. The spectrum of labeled proteins showed that approximately 6.1% of the fucosidases had a mass shift of 350 Da that fit with a single probe–protein adduct. Additionally, < 0.5% of the fucosidases had a mass shift that corresponded to two probe adducts on BfFucH (Appendix A).

To identify the specific amino acids on the fucosidase surface that were being labeled by the probe, both labeled and unlabeled samples were digested with pepsin at a low pH followed by LC-MS/MS analysis. Unfortunately, the resulting BfFucH samples did not yield good coverage for the adduct-labeled peptide fragments. Therefore, we repeated this analysis with the well-characterized bacterial GH29 TmFuc fucosidase for bottom-up analysis. We were now able to identify two unique peptide fragments that were labeled by the quinone methide from AH062.

To determine the exact site of the modification, tandem mass analysis was performed, whereby y*_n_* and b*_n_* were assigned based on the annotation proposed by Roepstorff and Fohlman (Figure 2A and Appendix A). These data revealed that the quinone methide generated from AH062 had labeled either tryptophan W80 or W105. In Figure 2B, the modification sites are mapped on the known crystal structure of TmFuc. The modifications on W80 and W105 are positioned outside the fucosidase active site, separated, respectively, by 27 Å and 28 Å from the active-site nucleophile D224. Although quinone methides have been reported to react with various nucleophiles, the one generated from AH062 shows a high preference for the surface tryptophans in TmFuc. The previous literature has shown a high selectivity of a very similar quinone methide for tryptophan residues in proteins [24]. A closer look at the tryptophan units of TmFuc showed that W80 and W105 are two tryptophans located further away from the active site compared to various other tryptophan residues (Appendix A), potentially indicating a slow generation of the quinone methide in AH062 that allows it to diffuse away from the active site. Additionally, sequence alignment with Clustal Omega on all three fucosidases used in this study showed that W80 is not conserved (Appendix A). However, all fucosidases shown in the alignment do possess an aromatic amino acid residue at the W105 position.

To further assess the utility of this probe for imaging bacterial fucosidase activity, we used AH062 to stain fucosidases associated with the cell wall of *B. fragilis*. Anaerobic overnight cultures of *B. fragilis* were incubated for 2h at 37 °C with AH062. As a negative control, bacterial samples were used that were preincubated with 100 µM of the competitive fucosidase inhibitor FNJ, followed by incubation with the probe. To visualize the probe, the bacteria were clicked by a copper(I)-catalyzed alkyne-azide cycloaddition (CuAAC) to an Alexa488 dye, and the bacterial cell membranes were stained with CellTrace^TM^ Yellow (CTY) to facilitate locating the bacteria (Figure 3A). Incubation with AH062 resulted in highly specific fluorescence labeling that was not observed after preincubation with FNJ. The AH062 pattern of labeling showed a similar localization on the outer membrane compared to the CTY cell membrane staining. By applying high-resolution structured illumination microscopy (SIM), we confirmed that AH062 stained the *B. fragilis* cell wall or cell membrane (Figure 3B). We next performed FACS analysis of AH062-labeled bacteria and observed a shift in the whole peak, indicating that most of the bacteria were fluorescently labeled by covalent attachment of the activated quinone methide. Again, labeling with AH062 could be blocked by preincubation with FNJ, showing the selectivity of the probe (Figure 4).

## 3. Materials and Methods

### 3.1. General Methods and Materials

All reagents and starting materials were obtained from commercial suppliers and were used without further purification. MeCN and CH_2_Cl_2_ and were dried and purified using an MBraun MB SPS 800 prior to use. Pyridine and DMSO were dried for 24 h over pre-activated (5 min, ~300 °C) 4Å molecular sieves prior to use. Glassware for anhydrous reactions was flame-dried and cooled under a nitrogen atmosphere immediately prior to use. Analytical TLC was performed on glass-backed TLC plates pre-coated with silica gel (60G, F_254_). Compound **10** was visualized by staining with molybdenum, ninhydrin, and 10% PPh_3_ in CH_2_Cl_2_ followed by ninhydrin. All other compounds were visualized under UV light and/or by staining with molybdenum and 5% sulfuric acid in EtOH. Column chromatography was performed with silica gel (230–400 mesh). ^1^H NMR, ^13^C NMR, and ^19^F NMR spectra were recorded on either a Bruker Avance 600 MHz NMR spectrometer or 400 MHz spectrometer in the designated deuterated solvents (CDCl_3_ or CD_3_OD). ^1^H and ^13^C NMR peak assignments were established based on ^1^H-^1^H COSY and ^1^H-^13^C HSQC experiments and, where possible, compared to previously reported data [19]. Chemical shifts (δ) are listed in ppm downfield from TMS using TMS as an internal reference. Coupling constants are reported in Hz. The following abbreviations were used to explain the multiplicities: s = singlet, d = doublet, t = triplet, q = quartet, m = multiplet, b = broad. Expression and purification of the recombinant proteins BfFucH and AfcA were based on previous publications [11,25]. Recombinant TmFuc and tissue-derived transferrin (TF), both of sufficient quality, were commercially available via, respectively, Megazyme (9037-65-4) and Calbiochem (616397).

### 3.2. Synthesis

#### 3.2.1. (2S,3S,4R,5R,6S)-2-bromo-6-methyltetrahydro-2H-pyran-3,4,5-triyl triacetate 3



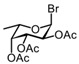



l-Fucose (1 g, 6.1 mmol) was dissolved in pyridine (6 mL) and Ac_2_O (6 mL), and the reaction mixture was stirred for 5 h at room temperature. Solvents were removed under reduced pressure, and the residue obtained was dissolved in 20 mL DCM. The solution was washed with 1M HCl (3 × 20 mL), H_2_O (3 × 20 mL), and brine (20 mL), dried over anhydrous Na_2_SO_4_, filtered, and concentrated under reduced pressure. Purification of per-acetylated L-fucose by silica gel column chromatography (eluent: 80% PetEt/EtOAc) provided the desired product in 99% yield as a colorless oil. R_f_ = 0.45 (PetEt/EtOAc = 2/1).

#### 3.2.2. (2R,3S,4R,5R,6S)-2-(2-formyl-4-nitrophenoxy)-6-methyltetrahydro-2H-pyran-3,4,5-triyl triacetate 5



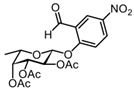



DIEA (2 mL, 12.33 mmol) was slowly added to an ice-cooled solution of crude bromide **3** (1.452 g, 4.11 mmol) and 2-hydroxy-5-nitrobenzaldehyde **4** (756 mg, 4.52 mmol) in anhydrous MeCN. The reaction mixture was stirred at room temperature for 24 h under argon. After completion of the reaction, the mixture was concentrated under reduced pressure. The residue was dissolved in EtOAc (50 mL), and the resulting solution was washed with 5% citric acid (3 × 25 mL), 5% NaHCO_3_ (3 × 25 mL), H_2_O (3 × 25 mL), and brine (25 mL). The organic phase was dried over anhydrous Na_2_SO_4_, filtered, and concentrated under reduced pressure which yielded a dark orange powder. Purification by silica gel column chromatography (eluent: 70% PetEt/EtOAc) provided the desired glycosylated product **5** (1.21 g, 2.75 mmol) as a white solid in 61% yield over 3 steps. R_f_ = 0.40 (PetEt/EtOAc = 6/4). ^1^H NMR (CDCl_3_, 400 MHz): δ 10.35 (s, 1H), 8.72 (d, *J* = 2.9 Hz, 1H), 8.43 (dd, *J* = 9.1, 2.8 Hz, 1H), 7.23 (s, 1H), 5.59 (dd, *J* = 10.5, 7.9 Hz, 1H), 5.36 (d, *J* = 3.4 Hz, 1H), 5.27 (d, *J* = 7.8 Hz, 1H), 5.17 (dd, *J* = 10.5, 3.4 Hz, 1H), 4.08 (q, *J* = 6.3 Hz, 1H), 2.22 (s, 3H), 2.06 (s, 3H), 2.03 (s, 3H), 1.31 (d, *J* = 6.4 Hz, 3H).

The ^1^H spectrum of this known compound was identical to a previously reported spectrum [19].

#### 3.2.3. (2S,3S,4R,5R,6S)-2-(2-formyl-4-nitrophenoxy)-6-methyltetrahydro-2H-pyran-3,4,5-triyl triacetate 6



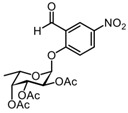



Anhydrous DMSO was added dropwise to a mixture of compound **5** (1.39 g, 3.16 mmol), a small amount of activated 3 Å molecular sieves (~3 beads), and K_2_CO_3_ (4.36 g, 31.6 mmol). The mixture was stirred for 48 h at room temperature under argon, diluted with 50 mL CHCl_3_, and washed with 5% NaHCO_3_ (3 × 25 mL). The combined aqueous layers were re-extracted with 25 mL CHCl_3_, and the combined organic phases were washed with 5% citric acid (2 × 25 mL) and brine (25 mL). The organic phase was dried over Na_2_SO_4_, filtered, and concentrated under reduced pressure which yielded a bright orange oil. Purification by silica gel column chromatography (eluent: 70% PetEt/EtOAc + 1% AcOH) provided the desired epimer **6** (0.68 g, 1.35 mmol) in 49% yield as a pale yellow solid, and the recovered yield for **5** (0.39 g, 0.88 mmol) was 28%. R_f_ = 0.73 (PetEt/EtOAc = 1/1). ^1^H NMR (CDCl_3_, 400 MHz): δ 10.53 (s, *J* = 1.5 Hz, 1H), 8.73 (d, *J* = 2.8 Hz, 1H), 8.44 (dd, *J* = 9.2, 2.9 Hz, 1H), 7.43 (d, *J* = 9.2 Hz, 1H), 5.94 (d, *J* = 3.5 Hz, 1H), 5.53 (dd, *J* = 11.0, 3.2 Hz, 1H), 5.42 (d, *J* = 3.2 Hz, 1H), 5.38 (dd, *J* = 10.9, 3.6 Hz, 1H), 4.27 (q, *J* = 6.4 Hz, 1H), 2.22 (s, 3H), 2.06 (s, 3H), 2.05 (s, 3H), 1.19 (d, *J* = 6.5 Hz, 3H).

The ^1^H spectrum of this known compound was identical to a previously reported spectrum [19].

#### 3.2.4. (2S,3S,4R,5R,6S)-2-(2-(hydroxymethyl)-4-nitrophenoxy)-6-methyltetrahydro-2H-pyran-3,4,5-triyl triacetate 7



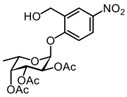



NaBH(OAc)_3_ (655 mg, 3.1 mmol) was added to a solution of compound **6** (388 mg, 0.88 mmol) in 8 mL EtOH. The reaction mixture was stirred for 24 h under nitrogen, diluted with EtOAc (100 mL), and washed with H_2_O (3 × 50 mL) and brine (50 mL). The organic phase was dried over Na_2_SO_4_, filtered, and concentrated under reduced pressure. Purification by silica gel column chromatography (eluent: 80% PetEt/EtOAc) provided the desired product **7** (311 mg, 0.71 mmol) in 81% yield as a white solid. R_f_ = 0.58 (PetEt/EtOAc = 1/1). ^1^H NMR (CDCl_3_, 400 MHz): δ 8.29 (d, *J* = 2.4 Hz, 1H), 8.17 (dd, *J* = 9.0, 2.5 Hz, 1H), 7.26 (d, *J* = 9.1 Hz, 1H), 5.81 (d, *J* = 3.6 Hz, 1H), 5.52 (dd, *J* = 10.9, 3.2 Hz, 1H), 5.43–5.35 (m, 2H), 4.89 (d, *J* = 13.8 Hz, 1H), 4.67 (dd, *J* = 13.8 Hz, 6.4 Hz, 1H), 4.25 (q, *J* = 6.5 Hz, 1H), 2.49 (b, 1H), 2.22 (s, 3H), 2.09 (s, 3H), 2.07 (s, 3H), 2.05 (s, 3H), 1.17 (d, *J* = 6.5 Hz, 3H).

The ^1^H spectrum of this known compound was identical to a previously reported spectrum [19].

#### 3.2.5. (2S,3S,4R,5R,6S)-2-(4-(tert-butoxycarbonylamino)-2-(hydroxymethyl)phenoxy)-6-methyltetrahydro-2H-pyran-3,4,5-triyl triacetate 8



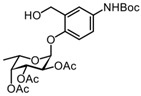



A Pd/C catalyst (10 wt. % Pd, 62 mg) was added to a solution of compound **7** (311 mg, 0.71 mmol) in EtOAc. The reaction mixture was purged and filled with H_2_ and then stirred for 30 min at room temperature. Thereafter, Boc_2_O (162 mg, 0.74 mmol) was added to the reactor, and the solution was stirred for 24 h at room temperature. More Boc_2_O was added, and the solution was again stirred for 24 h. The reaction mixture was filtered over a fritted glass funnel packed with Celite and concentrated under reduced pressure. The residue was diluted with 50 mL EtOAc and washed with H_2_O (3 × 25 mL) and brine (25 mL). The organic phase was dried over Na_2_SO_4_, filtered, and concentrated under reduced pressure. Purification by silica gel column chromatography (eluent: 70% PetEt/EtOAc + 1% AcOH) provided the desired product **8** (265 mg, 0.52 mmol) in 74% yield as a white solid. R_f_ = 0.71 (PetEt/EtOAc = 4/6). ^1^H NMR (CDCl_3_, 400 MHz): δ 7.30 (s, 1H), 7.23 (d, *J* = 8.8 Hz, 1H), 7.10 (d, *J* = 8.8 Hz, 1H), 6.38 (s, 1H), 5.58 (d, *J* = 3.6 Hz, 1H), 5.49 (dd, *J* = 10.9, 3.1 Hz, 1H), 5.42–5.33 (m, 2H), 4.85 (dd, *J* = 12.9, 4.7 Hz, 1H), 4.49 (dd, *J* = 12.9, 8.5 Hz, 1H), 4.33 (q, *J* = 6.4 Hz, 1H), 2.19 (s, 3H), 2.07 (s, 3H), 2.01 (s, 3H), 1.50 (s, 9H), 1.17 (d, *J* = 6.5 Hz, 3H).

The ^1^H spectrum of this known compound was identical to a previously reported spectrum [19].

#### 3.2.6. (2S,3S,4R,5R,6S)-2-(4-(tert-butoxycarbonylamino)-2-(fluoromethyl)phenoxy)-6-methyltetrahydro-2H-pyran-3,4,5-triyl triacetate 9



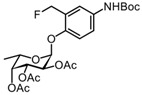



A solution of diethylaminosulfur trifluoride (DAST, 79 µL, 0.6 mmol) in anhydrous CH_2_Cl_2_ was added to an ice-cooled solution of compound 8 (204 mg, 0.40 mmol) in anhydrous CH_2_Cl_2_. The reaction mixture was allowed to warm up to room temperature and was stirred for 3.5 h under an argon atmosphere. The reaction was quenched by adding a small amount of silica and 10 drops of MeOH, followed by stirring for 10 min. Then, the reaction mixture was filtered over a fritted glass funnel and concentrated under reduced pressure. Purification by silica gel column chromatography (eluent: 80% PetEt/EtOAc) provided the desired product 9 (133 mg, 0.26 mmol) in 65% yield as a pale yellow oil. R_f_ = 0.78 (PetEt/EtOAc = 1/1). ^1^H NMR (CDCl_3_, 600 MHz): δ 7.42 (s, *J* = 15.1 Hz, 1H), 7.30 (d, *J* = 7.7 Hz, 1H), 7.10 (d, *J* = 8.8 Hz, 1H), 6.48 (s, 1H), 5.61 (d, *J* = 3.7 Hz, 1H), 5.55–5.32 (m, 4H), 5.28 (dd, *J* = 10.9, 3.7 Hz, 1H), 4.31 (q, *J* = 6.5 Hz, 1H), 2.20 (s, 3H), 2.06 (s, 3H), 2.03 (s, 3H), 1.51 (s, 9H), 1.16 (d, *J* = 6.5 Hz, 3H). ^13^C NMR (CDCl_3_, 151 MHz): δ 170.59 (C), 170.47 (C), 170.13 (C), 152.94 (C), 150.43 (d, *J* = 4.3 Hz, C), 133.50 (C), 126.53 (d, *J* = 16.8 Hz, C), 120.65 (CH), 120.05 (CH), 115.91 (CH), 96.19 (CH), 80.36 (C), 79.78 (d, *J* = 165.8 Hz, CH_2_F), 70.85 (CH), 67.97 (CH), 67.76 (CH), 65.50 (CH), 28.32 (CH_3_), 20.76 (CH_3_), 20.70 (CH_3_), 20.66 (CH_3_), 15.90 (CH_3_). ^19^F NMR (376 MHz, CDCl_3_): δ -212.13 (t, *J* = 52.1 Hz).

The ^1^H and ^13^C spectra of this known compound were identical to previously reported spectra [19].

#### 3.2.7. 3-(2-(2-(2-azidoethoxy)ethoxy)ethoxy)propanoic acid 10



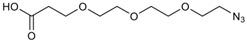



Imidazole-1-sulfonyl azide hydrochloride (231 mg, 1.1 mmol) was added to a solution of acid-PEG3-amine (204 mg, 0.92 mmol), K_2_CO_3_ (280 mg, 2.02 mmol), and CuSO_4_· 5 H_2_O (2.3 mg, 0.01 mmol) in MeOH. The reaction mixture was stirred for 6h at room temperature and concentrated under reduced pressure without heating. The residue was diluted with 15 mL H_2_O and acidified by dropwise addition of 1M HCl to pH 4. The mixture was extracted with EtOAc (3 × 10 mL), and the combined organic layers were dried over Na_2_SO_4_, filtered, and concentrated under reduced pressure. Purification by silica gel column chromatography (eluent: 60% PetEt/EtOAc) provided the desired product **10** (196 mg, 0.79 mmol) in 86% yield as a colorless oil. R_f_ = 0.18 (PetEt/EtOAc = 1/1). ^1^H NMR (600 MHz, CDCl_3_) δ 3.77 (t, *J* = 6.3 Hz, 2H), 3.72–3.59 (m, 10H), 3.40 (t, *J* = 4.9 Hz, 2H), 2.65 (t, *J* = 6.3 Hz, 2H). ^13^C NMR (151 MHz, CDCl_3_) δ 176.52 (CO_2_H), 70.66 (CH_2_), 70.63 (CH_2_), 70.45 (CH_2_), 70.41 (CH_2_), 70.02 (CH_2_), 66.30 (CH_2_), 50.66 (CH_2_N_3_), 34.82 (CH_2_). HRMS (ESI-quadrupole) calculated for C_9_H_17_N_3_O_5_ (M–H)^-^: 246.1095; found: 246.1096.

#### 3.2.8. (2S,3S,4R,5R,6S)-2-(4-(3-(2-(2-(2-azidoethoxy)ethoxy)ethoxy)propanamido)-2-(fluoromethyl)phenoxy)-6-methyltetrahydro-2H-pyran-3,4,5-triyl triacetate 11



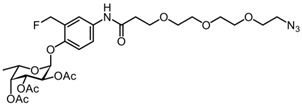



Compound **9** (156 mg, 0.304 mmol) was dissolved in 20% TFA/CH_2_Cl_2_, and the solution was stirred for 1.5 h at room temperature. The volatiles were concentrated under reduced pressure, and the flask was kept under high vacuum for 3 h to remove residual TFA. The residue obtained was dissolved in CH_2_Cl_2_, and compound **10** (83 mg, 0.336 mmol), 1-hydroxybenzotriazool (HOBt, 21 mg, 0.152 mmol), *N*,*N*-diisopropylethylamine (DIEA, 98 mg, 0.758 mmol), and 1-Ethyl-3-(3-dimethylaminopropyl)carbodiimide (EDC, 117 mg, 0.610 mmol) were added to the solution. The reaction mixture was stirred for 24h under nitrogen, diluted with EtOAc, and washed with 5% citric acid (3 × 25 mL), 5% NaHCO_3_ (3 × 25 mL), H_2_O (3 × 25 mL), and brine (25 mL). The organic phase was dried over anhydrous Na_2_SO_4_, filtered, and concentrated under reduced pressure. Purification by silica gel column chromatography (60% PetEt/EtOAc) provided the desired product **11** (178 mg, 0.277 mmol) in 91% yield as a pale yellow oil. R_f_ = 0.17 (PetEt/EtOAc = 1/1). ^1^H NMR (600 MHz, CDCl_3_): δ 8.65 (s, 1H), 7.59 (d, *J* = 8.1 Hz, 1H), 7.52 (s, 1H), 7.12 (d, *J* = 8.8 Hz, 1H), 5.64 (d, *J* = 3.2 Hz, 1H), 5.57–5.34 (m, 4H), 5.28 (dd, *J* = 10.9, 3.4 Hz, 1H), 4.31 (q, *J* = 6.4 Hz, 1H), 3.84–3.79 (m, 2H), 3.75–3.59 (m, 10H), 3.38–3.28 (m, 2H), 2.65 (m, 2H), 2.36 (s, 3H), 2.20 (s, 3H), 2.03 (s, 3H), 1.16 (d, *J* = 6.4 Hz, 3H). ^13^C NMR (151 MHz, CDCl_3_): δ 170.58 (C), 170.47 (C), 170.13 (d, *J* = 4.5 Hz, C), 133.48 (C), 122.32 (CH), 121.62 (d, *J* = 7.4 Hz, CH), 115.69 (CH), 96.14 (CH), 79.78 (d, 165.8 Hz, CH_2_F), 70.85 (CH), 70.67 (CH_2_), 70.52 (CH_2_), 70.37 (CH_2_), 70.25 (CH_2_), 70.01 (CH_2_), 67.98 (CH), 67.79 (CH), 67.03 (CH_2_), 65.54 (CH), 50.63 (CH_2_), 37.71 (CH_2_), 21.12 (CH_3_), 20.76 (CH_3_), 20.67 (CH_3_), 15.9 (CH_3_). ^19^F NMR (565 MHz, CDCl_3_): δ -212.23 (t, *J* = 47.7 Hz).

#### 3.2.9. 3-(2-(2-(2-azidoethoxy)ethoxy)ethoxy)-N-(3-(fluoromethyl)-4-(((2S,3S,4R,5S,6S)-3,4,5-trihydroxy-6-methyltetrahydro-2H-pyran-2-yl)oxy)phenyl)propanamide AH062 (12)



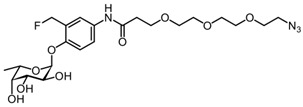



Na_2_CO_3_ (104 mg, 0.983 mmol) was added to a solution of compound **11** (158 mg, 0.246 mmol) in MeOH/CH_2_Cl_2_ (2/1), and the reaction mixture was stirred for 4 h. The reaction mixture was neutralized by H^+^ resin, filtered, and concentrated under reduced pressure. Purification by silica gel column chromatography (eluent: 99% to 94% PetEt/EtOAc gradient + 1% AcOH) provided the desired final product (**AH062;**
**12**) (88.8 mg, 0.172 mmol) in 70% yield as a colorless solid. R_f_ = 0.63 (CHCl_3_/MeOH = 8/2). ^1^H NMR (600 MHz, CD_3_OD): δ 7.62 (s, 1H), 7.53 (d, *J* = 9.2 Hz, 1H), 7.18 (d, *J* = 8.9 Hz, 1H), 5.56 (dd, *J* = 87.2, 11.2 Hz, 1H), 5.51 (d, *J* = 3.4 Hz, 1H), 5.48 (dd, *J* = 87.2, 11.2 Hz, 1H), 4.63 (s, NH), 4.06 (q, *J* = 6.6 Hz, 1H), 4.01–3.92 (m, 2H), 3.84 (t, *J* = 6.0 Hz, 2H), 3.76 (d, *J* = 2.3 Hz, 1H), 3.70–3.54 (m, 12H), 2.63 (t, *J* = 6.0 Hz, 2H), 1.20 (d, *J* = 6.6 Hz, 3H). ^13^C NMR (151 MHz, CD_3_OD): δ 170.85 (C), 132.76 (C), 121.54 (CH), 120.98 (CH), 114.92 (CH), 98.13 (CH), 79.85 (d, *J* = 164.1 Hz, CH_2_F), 72.06 (CH), 70.23 (CH_2_), 70.12 (CH), 70.09 (CH_2_), 69.68 (CH_2_), 68.27 (CH), 67.37 (CH), 66.80 (CH_2_), 37.08 (CH_2_), 15.22 (CH_3_). ^19^F NMR (565 MHz, CD_3_OD): δ -215.71 (t, *J* = 47.9 Hz). HRMS (ESI-quadrupole) calculated for C_22_H_33_FN_4_O_9_ (M + H)^+^: 517.2304; found: 517.2314.

### 3.3. Cloning, Expression, and Purification of Recombinant Fucosidases BfFucH and AfcA

The codon-optimized genes encoding for BbAfcA (amino acid 577-1474, GenBank: AY303700), including the addition of GGSGGSHHHHHHHH plus a stop codon, and BfFucH (amino acid, GenBank: CAH08937.1) were synthesized by Genscript and subcloned into pMAL-c4x-1-H(RBS)+ (AvaI/EcoRI) and pET47b+ (Acc65I/XhoI), respectively. BL21(DE3) (C2527H, NEB) cells were transformed with each of the individual vectors, and a colony was picked from the 2xYT agar (BP97432, Fisher Bioreagents, Pittsburgh, PA, USA) plate with the appropriate antibiotic (ampicillin 100 μg/mL, kanamycin 50 ug/mL, Cayman chemicals (Ann Arbor, Michigan, MI, USA) and expanded to a cell culture volume of 500 mL antibiotic (ampicillin 100 μg/mL, kanamycin 50 ug/mL) containing 2xYT medium at 37 °C. The cells were induced at OD600 = 0.6 with IPTG (final concentration was 1 mM, R0393, Thermo Scientific (Waltham, Massachusetts, MA, USA)) and cultured overnight at 20 °C. Then, cells were pelleted at 3000× *g*, resuspended in 5% of culture volume with lysis buffer (1 mg/mL lysozyme (62971, Sigma-Aldrich, St. Louis, MO, USA) and 0.1% triton X-100 (T8787, Sigma-Aldrich)), incubated at 37 °C for 1 h, and sonicated for 30 min on ice. A clear supernatant, by prior removal of cell debris by centrifugation at 10,000× *g*, was loaded onto a gravity flow 4 mL Ni-NTA column (17-5318-01, GE healthcare (Tokyo, Japan)). A standard buffer consisting of 50 mM TRIS-HCl and 250 mM NaCl pH 8 was created. Imidazole was added to the standard buffer at concentrations of 20 mM, 50 mM, or 250 mM to create wash 1, wash 2, and elution buffer (re-adjusted to pH 8 if needed), respectively. Once washed with 10 column volumes (CV) of wash 1, and 10 CV of wash 2, the enzyme of interest was eluted in 3 CV of elution buffer. The eluate was concentrated using a Vivaspin 6 filter (VS0602, Sartorius) and further purified on a PBS-equilibrated Superdex 200 Increase 10/300 GL column (28990944, GE healthcare) attached to a Shimadzu Nexera (Tokyo, Japan) system using a flow of 0.45 mL/min, collecting fractions each minute. Fractions containing a pure enzyme were pooled, concentrated, aliquoted, and stored at −20 °C in 10% glycerol in PBS upon further use.

### 3.4. Chemoselective Labeling

To study the limit of detection, various concentrations of BfFucH (4–0.25 µg) or AfcA (8–3.7 µg) were incubated with 350 μM of AH062 and then incubated for 1.5 h at 37 °C in PBS (pH 7.4). The negative controls included were a sample without any probe and a sample with the recombinant enzymes preincubated with the competitive inhibitor FNJ (final concentration 100 µM, 99212-30-3, Carbosynth, Berkshire, UK). The samples were mixed with 3× loading buffer, heated for 5 min at 95 °C, and loaded onto an SDS-PAGE gel. After an electrophoresis protocol of 10 min 90V, and 50 min 200V, the samples were blotted to a nitrocellulose membrane (TransBlot Turbo Transfer System, Biorad, Hercules, CA, USA). The membrane was washed for 2 min in PBS, after which it was incubated for 2 h at room temperature in the presence of 100 µM alkyne-biotin (1262681-31-1, Jena Bioscience, Jena, Germany), 2.5 mM L-ascorbate (134-03-2, Sigma-Aldrich), and 0.5 mM Cu_2_SO_4_ (7758-99-8, Sigma-Aldrich) in PBS. The blots were washed for 2 times 5 min with PBS containing 0.1% Tween-20 (PBS-T). The membrane was blocked using 5% skimmed milk in PBS-T with rocking for 1 h at room temperature. The blocking solution was decanted, and a solution of 1% skimmed milk in PBS-T containing α-biotin-HRP antibody conjugate (1:10.000, Jackson ImmunoResearch, West Grove, PA, USA) was added and incubated for 1h at room temperature. Membranes were washed for 3 × 10 min with PBS-T. Detection of membrane-bound αbiotin-HRP conjugates was accomplished by chemiluminescent detection using the Clarity Western ECL kit (Bio-Rad, Hercules, CA, USA) and imaged in a Gel-Doc system (Bio-Rad). A second gel with the same samples was run in parallel for each experiment, to which silver staining or PageBlue staining was applied to provide a loading control according to the protocol by Chevallet et al. [26].

### 3.5. Labeling Selectivity Assay

BfFucH (960 nM) was incubated with 350 µM AH062 in the presence of decreasing concentrations of TF (960–60 nM) for 2 h at room temperature in PBS. The enzyme mixtures were transferred, conjugated to alkyne-biotin, and visualized as described above.

### 3.6. Mass Spectrometry

For the peptide-centric LC-MS-MS, 5 µg expressed fucosidases, either labeled or unlabeled, were denatured and reduced using 5 mM tris(2-carboxyethyl)phospine (TCEP) and incubated for 15 min at 60 °C. Samples were subsequently alkylated with 20 mM chloroacetamide (CAA), digested proteolytically using pepsin (Roche) in a 0.2% trifluoroacetic acid (TFA) buffer at pH < 2 (1:75 enzyme/protein ratio), and incubated overnight at 37 °C. The protein was subjected to several proteases including trypsin, chymotrypsin, and glu-C (data not shown). Pepsin proved to be the most suitable for producing peptides at appropriate lengths with the highest sequence coverage. Following digestion, solid-phase extraction was performed using Oasis microElution 96-well plates (Waters, Wexford, Ireland). In short, wells were conditioned with acetonitrile (ACN) and then equilibrated with 0.1% TFA. The supernatant from the digestion was loaded and washed twice with 0.1% TFA. Peptides were eluted using 60% ACN 0.1 % TFA. Peptide eluates were dried by low-pressure rotary evaporation and resuspended in 2% formic acid for LC-MS^2^ analysis.

Per peptide sample, 50 ng was analyzed using a Thermo Scientific Ultimate HPLC nanoflow system, hyphenated to an Orbitrap Exploris 480 Mass Spectrometer (Thermo Fisher Scientific (Waltham, Massachusetts, MA, USA). Buffer A (0.1% formic acid (FA) in water) and buffer B (0.1% FA, 20% ACN in water) were used for peptide chromatography with the following gradient: 0 min 9% B; 2 min 13% B; 39 min 44% B; 42 min 99% B; 46 min 99% B; 47–55 min 9% B. Mass spectrometry analysis was performed in positive ion mode using electrospray ionization from a coated fused silica emitter at a 1900 V spray voltage. For MS^1^ scans, the mass range was set from *m/z* 375 to 1600 with a resolution of 60,000. The AGC target was set to the MS standard with automated maximum injection times. The data-dependent MS^2^ acquisition method initiated HCD fragmentation at 28% of normalized collision energy on the highest charge state and lowest *m/z* signals within a 1 s cycle time. The exclusion time was set to 10 s. The resolution for MS^2^ acquisition was at 15,000 with a mass range from *m/z* 40 to 2500. Here, the AGC target was also set to standard with automated maximum injection times. For data analysis of LC-MS^2^, we used the sequence of recombinantly expressed fucosidases from *T. maritima* and *B. fragilis*. Data were interpreted with Byonic v3.11.3 (Protein Metrics Inc.). Raw data were searched non-specifically with regard to digestion specificity. Precursor mass tolerance was set to 10 ppm, and HCD fragment mass tolerance to 20 ppm. As a fixed modification, Cys carbamidomethylation was included in the search. Met and Try oxidation was added as a common variable modification. The possible masses of the probe, protonated (351.17 Da) and deprotonated (350.16 Da), were searched against the known sequences of the enzyme (Appendix A).

For protein-centric LC-MS analysis, samples containing labeled and unlabeled fucosidases from *T. maritima* and *B. fragilis* were chromatographically separated on a Thermo Scientific Vanquish Flex UHPLC system. An amount of 8 µg of protein material was loaded on a reverse phase column heated to 80 °C. The LC-MS runtime was set to 22 min with a flow rate of 250 µL/min. Analytes were eluted using two mobile phases, namely, 0.1% FA in water (buffer A) and 0.1% FA in ACN (buffer B), with a gradient of 25–46% B for 14 min. The LC-MS analysis was performed on an Orbitrap Exploris 480 Mass Spectrometer expanded with intact (biopharma) mode. Data were collected with the instrument set to intact protein mode, at a resolution of 7500 at *m*/*z* 200. Raw spectra were charge state deconvoluted using PMI intact mass software at an *m*/*z* range of 500−2000, using a charge vector spacing of 0.6 with a baseline radius of an *m*/*z* of 15. Mass smoothing sigma was set to 3, and mass spacing to 0.5, and there was a maximum of 10 iterations. Annotations by the program were checked manually using the raw data.

### 3.7. Fluorescent Microscopy and Analysis

*B. fragilis* (NTCT9343) was grown anaerobically at 37 °C in GAM medium (Gifu Anaerobic Broth, 05422, HyServe). Overnight cultures (18 h) were collected via centrifugation (4000 rpm, 5 min, 4 °C) and resuspended in PBS, and 500 µL of OD_600_ 1.0 was transferred into aliquots. The aliquots were centrifuged (4000 rpm, 5 min, 4 °C) and resuspended in 100 µL PBS containing the competitive inhibitor FNJ (final = 100 µM) or an equal volume of PBS. Bacteria were incubated for 30 min at 37 °C. The appropriate samples were treated with 350 µM AH062 and incubated for 1.5 h at 37 °C. Bacteria were collected via centrifugation at 8000 rpm for 5 min and washed 2× with 1 mL PBS. Pellets were resuspended in 100 µL CuAAC reaction buffer (1 µM alkyne-Alexa488 (Jena Bioscience, Jena, Germany), 0.5 mM CuSO_4_, 2.5 mM l-ascorbate in PBS). Bacteria were incubated in the dark at RT for 3 h and collected by centrifugation as above. Bacteria were washed 3x by resuspension in PBS-T, incubated for 3 min in the dark, and centrifuged as above. The bacteria were resuspended in 50 uL PBS and labeled by 1 µL CTY (Cell trace yellow cell proliferation kit, Invitrogen) per vial for 20 min at 42 °C. The bacteria were washed 2× with PBS followed by fixation with 4% PFA for 20 min at RT. The bacteria were washed an additional 2 times in PBS. The bacterial pellets were resuspended in Prolonged Diamond solution (Thermo Fisher) and mounted onto cover slips for confocal imaging.

Images were either collected on a Leica SPE-II confocal microscope using a 100× objective (HCX PL APO CS 100.0 × 1.40 oil) controlled by Leica LAS AF software with default settings to detect Alexa488 and CTY, or on an OMX-V4 BLAZE super-resolution microscope using a 60× SIM lens (Olympus U-PLAN APO, NA 1.42, WD = 0.15). The raw SIM data were reconstructed in SoftWoRx (Cytiva), and channel alignment was performed using Chromagnon with reference images of calibration tetraspec beads (Thermo Scientific) [27]. Images were converted and deconvolved (blind, iteration 20×) in NIS elements (NIKON, Tokyo, Japan). Maximum intensity projections are shown.

### 3.8. Flow Cytometry

Bacteria were prepared similar to the procedure described for “Fluorescent Microscopy and Analysis” until the CTY labeling step. Pellets of clicked bacteria were fixed by the addition of 500 µL 4% PFA and incubated for 20 min at RT. Bacteria were washed 2× with PBS and taken up in 1 mL aquadest. Bacteria were diluted an additional 10× in aquadest and analyzed using a BD FACSVerse. Flow cytometry data were analyzed using FlowJo 10.

## 4. Conclusions

In summary, we report a novel fucosidase probe (AH062) that is capable of labeling both GH29 and GH95 bacterial fucosidases, thereby showing, for the first time, a probe that targets inverting fucosidases. The quinone methide generated by this probe mainly labels on specific tryptophan residues outside of the active site of the TmFuc fucosidase enzyme, with single protein–probe adducts dominating. The probe was successfully used for the detection of fucosidase-positive bacteria by high-resolution confocal microscopy and flow cytometry. For further application of this probe in complex co-culture systems, the likely occurrence of quinone methide labeling adducts on off-target proteins in co-cultures needs to be determined and minimized. In the future, we envision applying this novel probe for the labeling, identification, and characterization of important bacterial GH29 and GH95 fucosidases at the intestinal host–microbe interface.

## Data Availability

The data presented in this study are available on request from the corresponding author.

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
