# Peer review of "Detection of Bacterial α-l-Fucosidases with an Ortho-Quinone Methide-Based Probe and Mapping of the Probe-Protein Adducts"

_molecules, 2022, doi:10.3390/molecules27051615_

Round 1
Reviewer 1 Report
In the manuscript entitled "Detection of bacterial α-L-fucosidases with an ortho-quinone methide-based probe and mapping of the probe-protein adducts", the authors describe the development and testing of a novel probe for the detection of GH29/95 focusidases. In my opinion, the authors have presented their study in a very concise and scientifically accurate manner, and have given sound reasons for why their research is of importance to the study of human gut microbiomes. Aside from some small corrections (see below), I congratulate the authors for their work.
Corrections:
Lines 42-43: Missing refs.
Line 64: I think there should be a comma instead of a full stop after the refs.
Lines 68-73: Missing refs.
Lines 75-81; Missing refs.
Line 91: Missing full stop.
Line 185: I would suggest replacing the full stop with a comma instead.
Figure 1: I find this figure somewhat confusing, due to the lack of size labelling on the side of gels A and C, as well as the inconsistency in the labelling of the proteins and molecules used. I would suggest including a size label for each gel, and either using the abbreviated name or full name of the protein/molecule used on all the images. In addition, the abbreviation of FNJ is not described in the legend to the figure.
Reviewer 2 Report
The authors demonstrated in this manuscript that a novel fucosidase probe (AH062) is capable of labeling both 232 GH29 and GH95 bacterial fucosidases. This manuscript is well organized and discussed the novel method for labelling both 232 GH29 and GH95 bacterial fucosidases.
However, authors need to address the following concerns before publication.
- The authors mentioned that: “Fucosidases are associated with several pathological conditions and play an important role in the 21 health of the human gut”. Please give examples of related diseases.
- Even though Bacteroides Fragilis is an obligate anaerobe, Gram-negative bacteria native to the normal microbiome of the human colon, the authors describe its cultivation conditions as aerobic microorganism.
- Activity-based probes (ABPs) have been mentioned previously in full in abstract, but after this it needs to be abbreviated.
- Bacteroides fragilis is sometimes written in full and sometimes in abbreviation.
- For the first time, write T maritima in full.
Reviewer 3 Report
Luijkx, Y.M.C.A., et al. have reported an activity-based probe based on quinone methide for the bacterial α-L-fucosidases. The designed probe AH062 has a clickable azide handle instead of BODIPY reported by Hsu et al. A multidisciplinary approach that includes synthesis, mass spectrometry, and cellular studies has been adopted and results corroborate the conclusions of the study. I have only a few minor points to make in Scheme 2.
- Reaction time duration from compound 11 to 12 is written as 5 h in the Scheme 2, whereas in the experimental it is mentioned as 4 h.
- Similarly, from compound 9 to 11 (1 h versus 1. 5 h)
- Reaction time (24 h) for EDC reaction in scheme 2 to be included.
Reviewer 4 Report
The authors prepared an oxo-quinone derivative based probe for the detection of fucosidases. The labeling ability and selectivity were evaluated by Western blotting. The results showed that the probes were effective in labeling GH95 and GH29 phloxidases, especially the inverted phloxidase. The labeling efficiency and the amino acid sites were evaluated by tandem mass spectrometry, and the binding to tryptophan was confirmed. Using this labeling probe, they successfully have images of fucosidases-positive bacteria. The development of such probes using synthetic organic molecules is an important development in molecular biology. Therefore, this study matches the focus of this journal and the special issue. In addition, the experimental results of the evaluation of synthetic molecules and the functional evaluation of imaging are sound. Therefore, I believe that the paper is basically worthy of publication. However, it would be helpful for readers to evaluate the performance of this probe if the following points are revised.
About Evaluation of chemiluminescences (CL) intensity in Figure 1
Figure 1A shows low concentration dependence (value: 49, 50, 24, 35, 49, 4), while FIgure 1C shows high concentration dependence (value: 121, 86, 40, 8, 0). Is this a reflection of the difference between the GH95 and GH29 phloxidases? In other words, is this due to the selectivity of the labeled probe or to experimental error?
